# Ring-Modified Histidine-Containing Cationic Short Peptides Exhibit Anticryptococcal Activity by Cellular Disruption

**DOI:** 10.3390/molecules28010087

**Published:** 2022-12-22

**Authors:** Komal Sharma, Shams Aaghaz, Indresh Kumar Maurya, Shreya Singh, Shivaprakash M. Rudramurthy, Vinod Kumar, Kulbhushan Tikoo, Rahul Jain

**Affiliations:** 1Department of Medicinal Chemistry, National Institute of Pharmaceutical Education and Research, Sector 67, S.A.S. Nagar 160 062, Punjab, India; 2Center of Infectious Diseases, National Institute of Pharmaceutical Education and Research, Sector 67, S.A.S. Nagar 160 062, Punjab, India; 3Department of Medical Microbiology, Post Graduate Institute of Medical Education and Research, Sector 12, Chandigarh 160 012, India; 4Department of Pharmacology and Toxicology, National Institute of Pharmaceutical Education and Research, Sector 67, S.A.S. Nagar 160 062, Punjab, India

**Keywords:** iodinated histidines, membrane active peptides, anticryptococcal activity, iodopeptides, pore formation, cell lysis, antifungal agents

## Abstract

Delineation of clinical complications secondary to fungal infections, such as cryptococcal meningitis, and the concurrent emergence of multidrug resistance in large population subsets necessitates the need for the development of new classes of antifungals. Herein, we report a series of ring-modified histidine-containing short cationic peptides exhibiting anticryptococcal activity via membrane lysis. The *N*-1 position of histidine was benzylated, followed by iodination at the C-5 position via electrophilic iodination, and the dipeptides were obtained after coupling with tryptophan. In vitro analysis revealed that peptides Trp-His[1-(3,5-di-*tert*-butylbenzyl)-5-iodo]-OMe (**10d**, IC_50_ = 2.20 μg/mL; MIC = 4.01 μg/mL) and Trp-His[1-(2-iodophenyl)-5-iodo)]-OMe (**10o**, IC_50_ = 2.52 μg/mL; MIC = 4.59 μg/mL) exhibit promising antifungal activities against *C. neoformans*. When administered in combination with standard drug amphotericin B (Amp B), a significant synergism was observed, with 4- to 16-fold increase in the potencies of both peptides and Amp B. Electron microscopy analysis with SEM and TEM showed that the dipeptides primarily act via membrane disruption, leading to pore formation and causing cell lysis. After entering the cells, the peptides interact with the intracellular components as demonstrated by confocal laser scanning microscopy (CLSM).

## 1. Introduction

Multidrug resistance and a compromised immune system due to underlying diseases such as HIV, cancer and COVID-19 infection have rendered easily treatable fungal infections as life-threatening. Cryptococcal meningitis is one such condition, and caused 580,000 deaths in 2020, accounting for 19% of the AIDS-related deaths globally [1]. Various classes of drugs approved for the treatment of invasive fungal infections act on the fungal membrane (polyenes, azoles and echinocandins) and DNA (pyrimidines). For the treatment of cryptococcal meningitis, the first line of treatment includes induction, consolidation and maintenance therapy which primarily relies on the administration of amphotericin B, fluconazole and flucytosine in different combinations for varied time durations [2]. Moreover, various formulations of the most potent and commonly used drug amphotericin B have been developed over a period of time that have shown promising results as compared to the conventional therapy [3]. However, due to acute and chronic toxicities, such as nephrotoxicity associated with amphotericin B and flucytosine, neurotoxicity with fluconazole and various gastrointestinal complications, along with development of resistance in microbes due to extensive and reckless use of antifungal drugs, there is a need for the development of newer classes of antifungals [4]. Hence, various new chemical structural classes are being explored and different anticryptococcal agents are being developed. Some of them act via interactions with various components of the fungal cells such as inhibition of glycosyl phosphatidylinositol (fosmanogepix), fungal-selective calcineurin inhibitor (APX879) and fungal-selective Hsp90 inhibitor (resorcylate aminopyrazoles). However, anticryptococcal agents acting via membrane disruption and lysis are quite popular, for example peptides [5,6], benzothioureas, ibomycin and hydrazycins [7].

Antifungal peptides (AFPs) primarily act via membrane disruption; however, their mechanism may involve complex secondary actions, such as reactive oxygen species (ROS) generation, mitochondrial dysfunction, induction of apoptosis, cell cycle interruption and interaction with genetic material [8]. The membrane interacting properties of AFPs basically depend on a few structural characteristics of the peptide such as positive charge and hydrophobicity [9,10,11,12,13]. The choice of amino acids in a peptide sequence specifically determines its toxicity and pharmacokinetic profiles along with biological activities. Peptides with more hydrophobic residues may lead to hemolysis whereas hydrophilic amino acid containing peptides may not be able to pass through the cell membrane [14]. The advantages associated with peptides, such as less systemic toxicity and lower potential for the development of resistance, render them a promising alternative to small antibiotics [15]. However, certain limitations such as enzymatic degradation, short half-life and poor oral bioavailability can be easily circumvented using modified/unnatural amino acids [16,17,18] or backbone modification [19,20].

Various reports suggest that introduction of a halogen group imparts overall hydrophobicity to a molecule, apart from inducing stability and modulating the biological activity. Jia et al. demonstrated that introduction of a halogen on naturally occurring Jelleine-I enhanced the antimicrobial and anti-biofilm activity of the peptide, with iodinated Jelleine-I showing the strongest in vitro activity while its chloro and bromo counterparts showed the best in vivo efficacies [21]. Similarly, another study demonstrated that addition of a fluorine atom on the *C*-terminal amidated tritrpticin analogue, tritrp1, showed similar activities against *E. coli* when compared to the parent peptide [22]. Cruz et al. demonstrated that the replacement of chlorine with bromine produced a variant of lantibiotic NAI-107 with improved bactericidal activity against gram-positive pathogens [23]. On a similar note, Molchanova et al. demonstrated that incorporation of halogen atoms into otherwise inactive peptoids led to improved antimicrobial activities. However, the activities were strongly dependent on the choice of halogen atom, as chlorine or bromine led to higher activity than the parent peptoid while introduction of fluorine did not alter the activity [24]. In line with this, Gottler et al. reported that a hexafluoroleucine-containing variant of protegrin showed diminished activity against various bacterial strains [25]. Hence, the introduction of a halogen atom to peptides modulates the activities but the altered activities are largely dependent on the choice of halogen atom, type of peptide and the hydrophobic-hydrophilic balance obtained after the modifications.

Herein, we modulated the activity of a previously reported dipeptide **1** (Figure 1) [26] by incorporating an iodo group. In the previously reported series of Trp-His class of dipeptides, introduction of various substituted benzyl groups at the *N*-1 position led to the identification of an anticryptococcal peptide (**1**). The peptide **1** possesses a 3,5-di-*tert*-butylbenzyl group at the *N*-1 position of histidine and NHBzl group at the *C*-terminal providing hydrophobic character in the peptide. In the current study, we replaced the NHBzl with a methyl ester group and in an attempt to keep the hydrophobicity intact or balanced, an iodo group was introduced at the *C*-5 position of the histidine ring (**2**). In our recent work, we observed that the most active dipeptides possessed an NHBzl group at the *C*-terminal and free *N*-terminal. Further replacement of the *C*-terminal NHBzl with OMe and introduction of Boc at the *N*-terminal led to peptides with similar activities to that of the most active peptides. The slight difference in activities was dependent on the substitutions at the *N*-1 position of histidine [27]. Therefore, in the present work, a less bulky *C*-terminal, i.e., OMe was chosen, as a bulky halogen group was introduced at the *C*-5 position of histidine.

A series of dipeptides **2** (Figure 1) were synthesized by utilizing the backbone of peptide **1** and introducing an iodo group at the *C*-5 position of the histidine along with speculating the effect of various electron donating and electron withdrawing groups on the *N*-1 benzyl group in the modulation of antimicrobial activities.

## 2. Results and Discussion

### 2.1. Synthesis

***Synthesis of Boc-His* (*1-Bzl-5-iodo*)*-OMe*:** The *N*-terminal of His-OMe (**3**) was protected with a Boc group in the presence of di-*tert*-butyl dicarbonate (Boc_2_O) to obtain Boc-His-OMe (**4**, Figure 1) which was further subjected to iodination by direct electrophilic halogenation in the presence of *N*-iodosuccinimide at ambient temperatures under dark and inert conditions [28]. Under these conditions, both *C*-2,5-diiodo (**5**) and *C*-5-iodo (**6**) products were obtained. The crude products were purified using a silica gel column with hexane:ethyl acetate as a solvent system. Boc-His-5-iodo-OMe (**6**) was then benzylated at the *N*-1 position using various substituted benzyl bromides and the final derivatives (**7a–o**) were isolated in 72–92% yield (Figure 2).

***Synthesis of peptides* (*Series* 1–2):** Synthesis of the Trp-His(1-Bzl-5-iodo)-OMe class of peptides was carried out from the *C*- to the *N*-terminus by sequential deprotection and coupling reactions (Figure 2). Acidolysis of **7a–o** was accomplished with 6 M HCl to obtain **8a–o**, and then neutralized with DIEA.

The coupling of **8a–o** with Boc-Trp-OH was executed in the presence of a coupling cocktail combination of HOAt and DIC in DMF under microwave irradiation. The crude peptides were purified on a fully automated flash column chromatography system with solvent system of CH_2_Cl_2_:MeOH. Deprotection at the *N*-terminal of purified dipeptides (**9a–o**) was performed with methanolic HCl to obtain dipeptides **10a–o**. The purity of all the synthesized peptides was analyzed by HPLC on a LC-18 column with a run time of 40 min with a flow of 1 mL/min using a gradient system of 95–5% (A:B) where buffer A and B were 0.1% CF_3_COOH in H_2_O and CH_3_CN (see Appendix A).

### 2.2. Antimicrobial Susceptibility Testing

Peptides **9a–o** and **10a–o** were tested against selected strains of fungi and bacteria and the results are summarized in Table 1, Table 2 and Table 3. In Series 1, with all Boc-protected peptides **9a–o** (Table 1), moderate to low activities were observed against *C. neoformans*. Most of the peptides with moderately bulky groups such as **9a** (R = benzyl), **9c** (R = 4-*iso*-propyl), **9e** (R = 3-trifluoromethyl) and with halogenated benzyl groups **9i–9n** showed weak fungicidal effects (IC_50_ values ranging from 17.62 to 23.58 μg/mL) in comparison to the reference standard amphotericin B. The peptides **9a–o** were found to be inactive against all other fungal strains.

In Series 2, removal of a Boc group (Table 2) led to a significant increase in the anticryptococcal activity of the peptides. With an overall increase in the bulk from a benzyl group (**10a**, IC_50_ = 17.68 μg/mL) to 3,5-di-*tert*-butylbenzyl (**10d**, IC_50_ = 2.20 μg/mL, MIC = 4.01 μg/mL), a substantial increase in bioactivity was observed. Other peptides with electron donating groups such as **10b** (R = 4-*tert*-butyl) and **10c** (R = 4-*iso*-propyl) exhibited moderate activity with IC_50_ values of 5.15 μg/mL (MIC = 9.36 μg/mL) and 8.81 μg/mL (MIC = 16.02 μg/mL), respectively. Hence, it is concluded that in peptides with electron donating side chains, the activity decreased with the decrease in bulk in the order: 3,5-di-*tert*-butylbenzyl ˃ 4-*tert*-butyl ˃ 4-iso-propyl ˃ benzyl.

However, when the methyl group of the *tert*-butyl moiety was replaced with the less hydrophobic and highly electron withdrawing fluorine atom **10g** (IC_50_ = 13.53 μg/mL, MIC = 24.6 μg/mL), the activity was reduced by 2.6-fold. The position of the trifluoromethyl group on the benzyl ring also affected the activity, such that 3-CF_3_ (**10e**) was found to be moderately active with an IC_50_ value of 4.87 μg/mL (MIC = 8.86 μg/mL) whereas the 2-CF_3_ group containing peptide, **10f**, exhibited a much lower activity with an IC_50_ value of 17.70 μg/mL (MIC = 32.18 μg/mL). Hence, the activity decreased in order 3-CF_3_ ˃ 4-CF_3_ ˃ 2-CF_3_. The effect of the fluorine group was observed by direct introduction at different positions on the benzyl ring. The derivatives of 3,4-difluorobenzyl (**10i**) and 3,5-difluorobenzyl (**10j**) exhibited moderate activities with IC_50_ values of 10.28 μg/mL (MIC = 18.7 μg/mL) and 6.09 μg/mL (MIC = 11.08 μg/mL), respectively. However, the 3-fluorobenzyl derivative (**10k**) displayed a significant increase in activity with an IC_50_ value of 2.78 μg/mL (MIC = 4.64 μg/mL). This prompted us to evaluate various other 3-halobenzyl substituted derivatives. Peptides **10l**, **10m** and **10n** containing 3-chloro, 3-bromo, 3-iodo groups, respectively, were moderately active with IC_50_ values of 17.6, 9.23 and 17.6 μg/mL, respectively. However, when iodo was placed at the *ortho* position of the benzyl ring, a significant increase in activity was observed with an IC_50_ value of 2.59 μg/mL (MIC = 4.59 μg/mL). Furthermore, the incorporation of an electron withdrawing nitro group (**10h**) at the fourth position of the benzyl ring drastically reduced the activity of the peptide (IC_50_ < 20 μg/mL). In a nutshell, with an optimum balance of hydrophobicity, electronegativity and polarity, peptides tend to show enhanced activity. All the peptides were found to be inactive against *C. albicans*, *C. glabrata*, *C. parapsilosis* and *C. krusei* at the highest tested concentrations. Peptides with IC_50_ values less than 20 μg/mL from both series were further tested against bacterial strains (Table 3) but none showed very significant activity.

To summarize, peptides **10d** (R = 3,5-di-*tert*-butylbenzyl) and **10o** (R = 2-iodobenzyl) showed the best activities of the two series with IC_50_ values of 2.20 μg/mL (MIC = 4.01 μg/mL) and 2.59 μg/mL (MIC = 4.59 μg/mL), respectively. The presence of an iodo group at the *C*-5 position of the histidine in this series of dipeptides imparts the quintessential hydrophobicity, therefore aiding the fungicidal activity of the peptides. The presence of different halogen groups (Br, Cl and F) at *C*-5 and their impact on the biological activities will be further explored in future.

### 2.3. Cytotoxicity Assay

Cytotoxicity assays were carried out to determine the selectivity of the most active peptides **10d** and **10o** towards fungal cells and their toxicity towards mammalian cells. The peptides showed a non-cytotoxic nature against human cancer cells line (HeLa) and a non-cancerous mammalian cell line (HEK 293) at their MIC concentration and higher (Figure 3, Table 4).

### 2.4. Hemolytic Assay

Non-selective peptides do not differentiate between fungal cells and human erythrocytes and thereby interact with both, causing hemolysis. A hemolytic study was carried out to demonstrate the selectivity of the synthesized peptides towards fungal cells. Peptides **10d** and **10o** exhibited non-hemolytic behavior at their MIC values and above, as evident from the HC_10_ and HC_50_ values (Table 4); therefore, they show high selectivity towards *C. neoformans* cells.

### 2.5. Time Kill Kinetics

To determine the time-dependent fungicidal action of peptides **10d** and **10o**, their time–kill profile was studied. The assay was carried out with the active peptides **10d** and **10o**-treated cryptococcal cells while corresponding inactive counterparts **9d** and **9o**-treated cells along with untreated cells were taken as negative controls. Amphotericin B-treated cells were used as the positive control. The growth profile of fungal cells was determined for each sample over a period of 24 h.

The growth pattern of amphotericin B-treated cryptococcal cells showed a significant drop in colony forming units (CFUs) after 4 h. Peptide **10d** displayed significant fungicidal activity and inhibited the growth of cryptococcal cells after 12 h (Figure 4a). However, inactive peptide **9d**-treated cells did not show any growth inhibition and exhibited the growth curve similar to that of a standard curve shown by untreated cells. Furthermore, time–kill kinetics of the active peptide **10o** revealed a gradual decrease followed by complete inhibition of CFUs after 12 h (Figure 4b). However, negative peptide **9o**-treated and untreated cells showed similar growth curves. The results confirmed the fungicidal action of the tested peptides.

### 2.6. Drug Combination Study

The fungicidal nature of the active peptides was further evaluated when given in combination with clinical standards that are used for the treatment of invasive cryptococcal infections. Cryptococcal cells were treated with different concentrations of peptides **10d** and **10o** in combination with varying concentrations of amphotericin or fluconazole and the fractional inhibitory concentration (FIC) index was calculated (Table 5) [29].
(1)$$\mathrm{FIC}\;\mathrm{index}\;\left(\mathrm{FICI}\right)=\mathrm{FIC}\;\mathrm{of}\;A\;+\;\mathrm{FIC}\;\mathrm{of}\;B=\frac{{\mathrm{MIC}}_{Comb}^{A}}{{\mathrm{MIC}}_{alone}^{A}}+\frac{{\mathrm{MIC}}_{Comb}^{B}}{{\mathrm{MIC}}_{alone}^{B}}$$

FICI ≤ 0.5: synergistic; FICI > 4: antagonistic; and FICI 0.6–4.0: additive or indifferent.

Peptide **10d**, when used in combination with amphotericin B, exhibited a significant synergistic action with an FICI value of 0.18. The MIC value of peptide **10d** showed a 16-fold decrease from 4.01 to 0.25 μg/mL, whereas that of amphotericin B showed an 8-fold decrease in the MIC value. Furthermore, **10d** also exhibited synergism with fluconazole, showing an FICI value of 0.19. An approx. 8-fold decrease in the MIC of **10d** (from 4.01 to 0.501 μg/mL) was observed and fluconazole showed an approx. 16-fold decrease in the MIC value (from 10 to 0.625 μg/mL). Peptide **10o**, at a non-cidal concentration, resulted in the 4-fold potentiation of amphotericin B in combination, while the MIC of peptide **10o** was lowered by around 16-fold (MIC from 4.59 to 0.285 μg/mL). The FICI value of this combination was found to be 0.31, indicating synergistic action. However, with fluconazole, peptide **10o** exhibited an additive effect with an FICI value of 1.0.

### 2.7. Mechanistic Investigations

The mechanism of action of the bioactive peptides was determined by analyzing the cell death pattern, by establishing the morphological changes after the treatment using electron microscopy and by identifying the broad intracellular targets using confocal imaging.

#### 2.7.1. Cell Death Pattern Analysis Using Flow Cytometry

Apoptosis includes a series of events occurring inside the cell, causing changes in plasma membrane, genetic material and various internal proteins, therefore leading to programmed cell death. One of the early markers of apoptosis is the translocation of a membrane phospholipid, phosphatidylserine, from the inner to the outer leaflet of the plasma membrane without altering the integrity of the plasma membrane [30]. Annexin V-FITC interacts with the externalized phosphatidylserine, therefore showing the cell change towards apoptosis which can be analyzed using flow cytometry [31,32]. The four quadrants in flow cytometry indicate different stages of the cells. The percentage depicts the population of the cells in a specific stage. Q1 corresponds to the percentage of cells undergoing necrosis, Q2 depicts the early apoptotic stage of the cells whereas Q4 corresponds to the cells in late-stage apoptosis and Q3 depicts the cells in the native state [33].

The control samples for peptide **10o** showed cryptococcal cells alone, exhibiting 100% population in native state Q3 (Figure 5a), whereas PI incubated samples (Figure 5b), showed 99.3% of cell population in the native state and 0.7% were in Q1. After incubation with annexin V-FITC (Figure 5c), the population of cryptococcal cells scattered more towards Q4 (6.4%), while 93.6% were in native quadrant Q3. Further incubation of cells with both PI and annexin V-FITC (Figure 5d) did not have a significant effect on the redistribution of the cell population over all quadrants, as more than 95% cells were in native state Q3, Q1 had 0.6% cells, Q2 showed 0.1% and Q4 constituted 3.3% of total cell population. However, when peptide **10d**-treated fungal cells were incubated with both PI and annexin V-FITC (Figure 5e), a marked variation in the pattern of distribution of cells was observed over the four quadrants, as 52.2% of cell population was pushed towards the Q4 quadrant showing late-stage apoptosis, while 1.7% cells exhibited necrosis (Q1), 1.3% were in Q2 and 44.8% of cell population remained in native state Q3. On a similar note, when cells were treated with **10o** and incubated with PI and annexin V-FITC (Figure 5f), 2% necrosis was observed (Q1), 48.1% exhibited late stage apoptosis and only 1% were in Q2, whereas 48.9% were in the native state Q3.

#### 2.7.2. Surface Morphology Analysis Using Scanning Electron Microscopy (SEM)

A comparative analysis of the surface morphologies of treated and untreated cells was carried out using scanning electron microscopy (SEM). The untreated cells appeared as oval-shaped with smooth surfaces and intact cell membranes (Figure 6a–c) when observed in SEM. However, cells treated with peptide **10d** showed large pores in the middle of the cells (Figure 6d–f) giving a doughnut-shaped appearance. The cell surface comparatively became highly wrinkled and a blatant change in morphology of the cells was observed in comparison to the untreated cells. Likewise, cryptococcal cells treated with peptide **10o** showed a highly wrinkled cell surface with pores in the middle making their overall appearance doughnut-like (Figure 6g–i) when compared to the untreated cells. The cells were highly deformed and the intracellular contents appear to have come outside the cell, thereby causing cell death.

#### 2.7.3. Internal Cell Morphology Analysis Using Transmission Electron Microscopy (TEM)

High-resolution transmission electron microscopy (HRTEM) provided an explicit view of the structural details of cells along with intricate changes that occur during the cell wall disruption. The cryptococcal cells in Figure 7a–c display a distinct inner plasma membrane covered with a layer of outer membrane followed by a faded capsule. The cell organelles can also be distinguished faintly from the images. However, the cells treated with peptide **10d** (Figure 7d–f) show complete lysis of the cell membrane. The outer membrane and capsule are not clearly distinguishable. The intracellular content of cells have oozed out and the cells appear as a stack of non-distinguishable structures lying one over the other. However, the level of cell membrane disruption was much more severe in the case of the 10o-treated cells (Figure 7g–i). The cells lost their shape completely and appear to have fused intracellular material. The cells were highly shrunken and the cell membrane was completely damaged as compared to the untreated cells.

Scanning transmission electron microscopy (STEM) analysis provides high contrast images of the cells and the results were in coherence with the HRTEM. The STEM images of untreated cells (Figure 8a,b) depict smooth walled, rounded cells with cell organelles intact, whereas the images of cells treated with peptide **10d** (Figure 8c,d) show highly shrunken cells with irregular cell membranes. The intracellular material has oozed out and a clear distinct cell was not observed. The deformed cells have merged to appear similar to a mass of highly irregular structure. Similarly, cryptococcal cells treated with **10o** (Figure 8e,f) were highly deformed and shrunken. The membrane separating the cells seemed to have disappeared and merged to form a cluster of deformed cells.

#### 2.7.4. Confocal Laser Scanning Microscopy (CLSM)

(a)Membrane permeabilization and localization of FITC-labeled peptide

The bioactive peptides **10d** and **10o** along with an inactive peptide **10h** were labeled with FITC (fluorescein isothiocyanate, a green, fluorescent dye, with excitation at 480 nm). The permeabilization of peptides inside the cell was analyzed by incubating the FITC-labeled peptides with *C. neoformans* cells and observing the incubated slides under CLSM. The FITC-tagged inactive peptide, on incubation with cryptococcal cells, did not show any fluorescence when observed under CLSM (Figure 9a), depicting their inability to cross the cell membrane. However, the images of cells incubated with FITC-tagged peptides **10d** (Figure 9b) and **10o** (Figure 9c) showed prominent green fluorescence. Hence, it was concluded that peptides act via membrane permeabilization; therefore, the membrane integrity of the cells is lost leading to permeabilization of the FITC-peptide inside the cells followed by localization, providing a bright green fluorescence.

(b)Membrane disruption and DNA interaction analysis by propidium iodide (PI) uptake

Propidium iodide (PI) dye cannot permeate the walls of healthy cells; hence, when cryptococcal cells were incubated with PI and observed under CLSM (excitation at 480 nm), they did not show any fluorescence (Figure 9d). However, when the cells were treated with peptide **10d** (Figure 9e) and **10o** (Figure 9f) and then incubated with PI, a prominent red fluorescence was observed. The results indicated that the peptides acted by lysis of the cell wall, which allowed the dye to penetrate the cell and intercalate with the DNA giving the whole cell a bright red fluorescence.

(c)Detection of nuclear fragmentation

The molecule 4′,6-diamidino-2-phenylindole (DAPI) easily permeates inside cells and interacts with the DNA giving a pinpoint blue fluorescence when observed under CLSM. It selectively binds to the minor groove of the DNA; therefore, any change in the fluorescence intensity pattern reveals the degree of damage to DNA. Untreated cells, when incubated with DAPI, showed faint blue pinpoint fluorescence (Figure 9g) indicating staining of the nucleus (marked by green arrows). However, cells treated with **10d** (Figure 9h) and **10o** (Figure 9i) exhibited fragmented bright blue fluorescence showing destroyed cells (marked by yellow arrows). Therefore, it can be concluded that after entering the cells, the peptides interact with the genetic material, triggering the event of cell death.

#### 2.7.5. Possible Mechanism

Primarily, the negatively charged surface of fungi demands a positively charged moiety for easy interaction; hence, an overall positive charge was introduced in the peptides while designing them by incorporating charged amino acids. Upon interaction with the fungal cell membrane, the hydrophobicity of peptides plays a crucial role and helps the peptide to permeate inside the lipophilic membrane of the fungal cell. During permeation, the peptide induces morphological changes in the otherwise intact fungal cell membrane leading to the appearance of wrinkles on the cell surface, as evident from SEM. The peptide then interacts with various intracellular components inside the cell leading to depletion of cytoplasm (TEM), and triggering various apoptotic pathways as confirmed by flow cytometry. The depletion leads to an altered morphology of the cell, making it appear doughnut-shaped (SEM). Consequently, the intact membrane become indistinguishable from cytoplasm (TEM). The collective effect of different interactions result in distortion of the cell surface integrity, membrane permeabilization, pore formation, cell lysis and death.

## 3. Conclusions and Summary

In a series of *N*-substituted histidine-containing His-Trp class of dipeptides, we found that an additional substitution at the C-5 position with an iodo group influenced the biological activity against *C. neoformans*. In vitro antifungal susceptibility assays revealed that peptides Trp-His[1-(3,5-di-*tert*-butylbenzyl)-5-iodo]-OMe (**10d**) and Trp-His[1-(2-iodophenyl)-5-iodo)]-OMe (**10o**) exhibited the most significant antifungal activities against *C. neoformans* among the whole series and showed high selectivity and safety profiles. The mechanistic studies revealed that the peptides are membrane active in nature as evident from SEM and TEM analyses, which demonstrated the contrasting morphologies of healthy and bioactive peptide-treated cells. Perfectly intact oval-shaped cryptococcal cells attained a highly deformed structure with a large pore in the middle which gave it a doughnut-like appearance. Furthermore, CLSM analysis showed that the peptides easily moved across the cell membrane by increasing fluidity, thereby causing membrane lysis and disruption and interaction with various intracellular targets expediting the cell death. Furthermore, flow cytometry showed that the cell death was hastened by the induction of apoptotic pathways.

## 4. Material and Methods

*Microbial strains:* Microbial strains were procured from National Culture Collection of Pathogenic Fungi (NCCPF) at the Postgraduate Institute of Medical Education & Research (PGIMER), Chandigarh and Microbial Type Culture Collection and Gene bank (MTCC) at the Institute of Microbial Technology (IMTECH), Chandigarh, India. Peptides were screened against five different strains of fungi: *viz. Candida albicans* (NCCPF400034), *Candida glabrata* (MTCC 3019), *Candida parapsilosis* (NCCPF 440002), *Candida krusei* (NCCPF 440002) and *Cryptococcus neoformans* (NCCPF250316), and five strains of bacteria: *viz. Escherichia coli* (MTCC 2961), *Staphylococcus aureus* (MTCC 3160), *Enterococcus faecalis* (MTCC 439), *Streptococcus pyogenes* (MTCC 442) and *Pseudomonas aeruginosa* (MTCC 3542).

*Growth media:* Antifungal susceptibility testing was carried out using standard broth microdilution assays as per the Clinical and Laboratory Standards Institute (CLSI) guidelines for fungi [34,35]. RPMI 1640 (Sigma), yeast extract, peptone and dextrose (YPED, Himedia) were used as the incubation broths. The cells were diluted to a cell density of 10^3^ colony forming unit (CFU/mL) and incubated at 30 °C for 48 h. Furthermore, Muller Hinton (Himedia) media was utilized for bacterial strains and antibacterial activity was evaluated per CLSI guidelines for bacteria [36]. Bacterial cells were diluted to a cell density of 10^5^ CFU/mL and incubated at 37 °C for 24 h. Amphotericin B (Himedia, India) and rifampicin (Himedia, India) were used as standard drugs for the evaluation of activity against all the fungal and bacterial strains, respectively.

For the detailed synthetic procedures and characterization data please refer to Appendix A.

### 4.1. Broth Microdilution Assay

The stock solutions of peptides (2 mg/mL) were prepared in DMSO. Specific amounts of peptide from stock solution were added to a 96-well microtiter plate containing 100 μL of growth media and then serially diluted two-fold. A further 100 μL of growth media containing microbial cells were added to each well and incubated at 30–37 °C for 24–48 h. The optical density (OD) was measured at 600 nm using a BioStack Ready (BioTek Instruments, Winooski, VT, USA) microplate reader to analyze microbial growth inhibition, and IC_50_ and MIC values were calculated. To visualize the growth and inhibition pattern, a solution of 3-(4,5-dimethylthiazol-2-yl)-2,5-diphenyltetrazolium bromide (MTT) in water was added to the plate and incubated for 2–3 h. MTT reacts with the mitochondrial dehydrogenases, which are present only in a viable cell, to give blue colored formazan crystals. Therefore, a darker color indicates a high number of viable cells; however, non-viable cells appear pale or colorless.

### 4.2. Cytotoxicity Assay

The cytotoxicity studies of peptides **10d** and **10o** were carried out in standard 96-well cell culture plates against HEK-293 and HeLa cells. The cells (50,000 cells/well) were seeded to the wells of the plate in Dulbecco’s Modified Eagle Medium (DMEM) supplemented with 10% fetal bovine serum at 37 °C overnight and incubated for 24 h to achieve confluency. Peptides at different concentrations of 50, 25, 12.5, 6.25, 3.12, 1.56 and 0.78 μg/mL were added to the cells in separate wells and incubated at 37 °C for 18 h. A further 20 µL MTT solution (5 mg/mL stock conc. in PBS) was added and the plate was incubated for another 3–4 h. To detect the viable cells that react with MTT to make formazan crystals, 120 μL of supernatant was removed and 100 μL DMSO was added to dissolve the crystals. Untreated cells and DMSO (10%) were used as negative and positive controls, respectively [37]. The percent viability of cells was calculated using equation:(2)$$\%\;\mathrm{Cell}\;\mathrm{viability}=\frac{\mathrm{Av}.\;\mathrm{Absorbance}\;\mathrm{of}\;\mathrm{treated}\;\mathrm{cells}}{\mathrm{Av}.\;\mathrm{Absorbance}\;\mathrm{of}\;\mathrm{control}\;\mathrm{cells}}\times 100$$

### 4.3. Hemolytic Assay

Human blood in 10% citrate phosphate dextrose was obtained from Government Hospital, Chandigarh, India. Red blood cells (RBCs) were harvested by spinning at 1000× *g* for 5 min followed by washing with PBS 3–5 times. The packed cell volume was obtained and utilized to make a 0.8% (*v*/*v*) suspension of RBCs in PBS. The suspension (100 μL) was transferred to each well of a 96-well microtiter plate and mixed with peptides **10d** and **10o** (100 μL) at concentrations of 50, 25, 12.5, 6.25, 3.12, 1.56 and 0.78 μg/mL. The plate was incubated at 37 °C for 60 min followed by centrifugation at 1000× *g* for 5 min at 25 °C. The supernatant (100 μL) was transferred to a fresh microtiter plate and absorbance was recorded at 414 nm (Varioskan LUX multimode microplate reader, Thermo Scientific, Waltham, MS, USA) to analyze RBC lysis. Untreated RBCs in PBS and lysed RBCs in 0.1% Triton X-100 were used as negative and positive controls, respectively [29]. Percent hemolysis was calculated using following equation [38].
(3)$$\%\;\mathrm{Hemolysis}=\frac{\mathrm{Absorbance}\;\mathrm{of}\;\mathrm{treated}\;\mathrm{sample}-\mathrm{Absorbance}\;\mathrm{of}\;\mathrm{negative}\;\mathrm{control}}{\mathrm{Absorbance}\;\mathrm{of}\;\mathrm{positive}\;\mathrm{control}-\mathrm{Absorbance}\;\mathrm{of}\;\mathrm{negative}\;\mathrm{control}}\times 100$$

### 4.4. Time Kill Assay

*C. neoformans* cells (~1 × 10^3^ CFU/mL) were inoculated in media and four samples for each test peptide were prepared, where the cryptococcal cells were either (i) incubated with test peptides (**10d** or **10o**), (ii) negative peptides (**9d** or **9o**), (iii) amphotericin B or (iv) left untreated in separate tubes with incubation broth. The tubes were then incubated for 24 h at 30 °C with 150× *g* in a shaking incubator. Aliquots of 100 μL were removed from each sample at predetermined time intervals of 0, 4, 8, 12, 16, 20 and 24 h. Aliquots of 10 μL were serially diluted (10-fold) in saline and plated on nutrient agar plates. The plates were then incubated at 30 °C for 24 h and the colony forming units (CFUs) of *C. neoformans* cells were counted [29].

### 4.5. Drug Combination Study

Various dilutions of standard drugs, i.e., amphotericin B, fluconazole and bioactive peptides (**10d** and **10o**) were prepared. Incubation broth was added to 96-well microplates, followed by addition of different concentrations of the standard drugs across the rows, and different concentrations of bioactive peptides across the columns of the microtiter plates. Hence, a unique combination of concentrations of standard drug and test peptide were obtained in each well. Inocula were added to the microplate by correcting the OD600 of fungal suspension in incubation broth. The plate was then incubated for 48–72 h at 30 °C. The optical density of each well of the plate was recorded on an ELISA plate reader to calculate the FIC index.

### 4.6. Flow Cytometry

*C. neoformans* cells were treated with the bioactive peptides (**10d** or **10o**) at MIC value and incubated at 30 °C at 200× *g* for 12 h. The treated cells along with a batch of untreated cells were centrifuged at 2000× *g* for 10 min at 4 °C and suspended in cold phosphate-buffered saline (PBS), followed by 2–3 washings with PBS. The harvested treated and untreated cells were then incubated with zymolyase for 1 h at ambient temperature to obtain cell protoplasm. The staining of obtained protoplasm was done using an Annexin V-FITC detection kit (Sigma-Aldrich, Saint Louis, DE, USA). The protoplasm of untreated cells was divided into 4 tubes (2 mL) and into 3 tubes a different staining agent was added while one was kept unstained. Hence, four samples of untreated cells were prepared with (i) unstained cells alone, (ii) cells with propidium iodide (PI), (iii) cells with annexin V-FITC and (iv) cells with both PI and annexin V-FITC. Furthermore, two more samples were prepared where protoplasm of **10d**- and **10o**-treated cells were stained with both PI and annexin V-FITC. All 6 samples were then analyzed by using an FACS Calibur (Becton-Dickinson, San Jose, CA, USA) at excitation of 488 nm (for FITC) and 560 nm (for PI) band filter. Approximately 10,000 cells were used for counting [29].

### 4.7. SEM

The bioactive peptides (**10d** or **10o**) were added at their MIC value to a suspension of *C. neoformans* cells (~10^3^) in an incubation broth at 30 °C for 12 h. Then, the treated, as well as untreated, cells were centrifuged separately at 2000× *g* for 10 min at 4 °C and suspended in cold PBS and followed by 2–3 washings with PBS. The cells were fixed with 2% glutaraldehyde in 0.1% phosphate buffer for 1 h at ambient temperature and washed 2–3 times with PBS. An amount of 1% osmium tetroxide in 0.1 M PBS was added to the cells to further fix and stain them. The cells were incubated with OsO_4_ for 1 h at 4 °C. The samples were then dehydrated by incubating them with increasing gradients of 10%, 30%, 50% and 70% of ethanol in water followed by 100% ethanol. The samples were placed on a round glass cover slip previously attached to the aluminum stub by adhesive carbon tape. The samples were dried using a vacuum freeze-drier HITACHI 2030 at ambient temperature, then sputter-coated with gold using a HITACHI E1010 and observed under the SEM (Hitachi S-3400N, Tokyo, Japan) [39].

### 4.8. TEM

The samples were prepared in a similar manner to as mentioned in the procedure for SEM sample preparation, up until the dehydration of cells with increasing gradients of ethanol. The dehydrated samples were then embedded in resin which was prepared from an epoxy embedding medium kit (Sigma, 45359-1EA-F). Increasing gradients (10%, 20%, 30%, 50%, 70% and 90%) of resin were added to the samples followed by incubation for 1–2 h at room temperature after addition of each gradient. Undiluted resin (100%) was added to the sample and incubated for 24 h at 70 °C. Semi-thin and ultrathin sections were cut with an ultramicrotome (Ultramicotome Lecia EM UC6). The sections of samples were placed on a 3.05 mm diameter and 200 mesh copper grid and stained with uranyl acetate or lead acetate, before observation under the HRTEM (FEI Technai G2 F20, Eindhoven, The Netherlands) and analysis at 120 KV.

### 4.9. CLSM

(a)FITC-labeled peptide uptake assay

Active and inactive peptides tagged with fluorescein isothiocyanate were incubated with *C. neoformans* cells (~1 × 10^3^ CFU/mL) at their MIC for 12 h. After incubation, cells were harvested by centrifugation at 2000× *g* for 10 min at 4 °C and suspended in PBS and mounted on a slide for visualization under a confocal microscope with excitation and emission wavelengths of 488 nm and 515 nm, respectively.

(b)PI uptake assay

The samples for PI were prepared by incubating *C. neoformans* (ATCC350) cells (~1 × 10^3^ CFU/mL) with test peptides (**10d** and **10o**) for 12 h at their MIC values. After incubation, treated and untreated cells were centrifuged at 2000× *g* for 10 min at 4 °C and suspended in cold PBS and followed by 2–3 washings with PBS. The cells were then incubated with PI (1.49 μM) for 1 h at room temperature with constant shaking (150× *g* rpm), and later harvested by centrifugation and suspended in PBS. Then, the cells were examined using confocal microscopy (Olympus Fluoview^™^ FV1000 SPD; Olympus, Tokyo, Japan) with a wavelength of >560 nm for PI. *C. neoformans* cells without treatment of active peptides served as a control.

(c)DAPI uptake assay

*C. neoformans* (ATCC350) cells (~1 × 10^3^ CFU/mL) were incubated with test peptides for 12 h at their MIC values. After incubation, treated and untreated cells were centrifuged at 2000× *g* for 10 min at 4 °C and suspended in cold PBS and followed by 2–3 washings with PBS. The cells were then incubated with DAPI (3.01 μM) for 10 min at room temperature with constant shaking, harvested by centrifugation and suspended in PBS. Then, the fungal cells were examined using confocal microscopy (Olympus Fluoview^™^ FV1000 SPD; Olympus, Tokyo, Japan) with excitation at 350 nm and emission at 470 nm [40].

## Data Availability

The data will be made available by the corresponding author on reasonable request.

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
