# Peer review of "Ring-Modified Histidine-Containing Cationic Short Peptides Exhibit Anticryptococcal Activity by Cellular Disruption"

_molecules, 2022, doi:10.3390/molecules28010087_

Round 1
Reviewer 1 Report
The authors present a series of ring-modified histidine containing short cationic peptides exhibiting anticryptococcal activity via membrane lysis. They study in deep two of them. With electron microscopy experiments they propose that the peptides primarily act via membrane disruption leading to pore formation causing cell lysis, and that after entering the cells, the peptides interact with the intracellular components as demonstrated by CLSM studies.
I cannot evaluate the synthesis of the peptides and purification stages, it is not my expertise. I will focus on the experiments related to the antifungal properties of the peptides.
The activity of all peptides against the tested fungi is rather low compared to Amphotericin B. This fact makes the work less interesting since it focuses on the study of these peptides as antifungal, and has to be clear in the abstract and introduction.
The combined effect of two of them with Amphotericin B is promising, and focus should be there. In this regards, cytotoxicity and hemolitic assays with Amphotericin B + 10 d or 10 o are mandatory.
Regarding the mechanistic studies:
I do not agree with the conclusion from the electron microscopy images, the presence of peptide-induced pores cannot be affirmed from these images. Cells appear as donuts, with membrane in the interior of the hole, not a membrane pore. Moreover, experiments with FITC labeled peptides show the peptides inside the cells and not in the membrane region. Therefore, it is not clear that they act at the membrane level.
Minor corrections:
CLSM is undefined in the abstract. Avoid abbreviates in abstract
Define R in each scheme
Add errors in the tables.
The sentence “One of the early markers of apoptosis is translocation of a membrane protein, phosphatidylserine from inner to the outer leaflet of the plasma membrane without altering the integrity of the plasma membrane ” is unclear. PS is a lipid, not a protein
Fig. 4: which concentration was used?
What means C. neoformans in the leyend? No addition of peptide?
Reviewer 2 Report
In this report the authors have reported a series of ring modified histidine containing short cationic peptides exhibiting anticryptococcal activity. They have tried to influence the membrane disruption effect of these peptides by introducing varieties of lipophilic groups in the histidine ring. The authors did a good job with the mechanistic investigations on two active peptides by SEM and TEM analysis in addition to other microbiological assays.
My biggest objection to this manuscript is the restricted Structure Activity Relationship (SAR ) studies that dampens my enthusiasm to recommend this manuscript for a straightforward acceptance without further work.
There are several instances where I was unable to understand the rationale of analog designing such as:
1) rationale of replacing the NHBzl with methyl ester group to keep the hydrophobicity intact or balanced-how? Please elaborate!
2) the choice of iodo group introduced at the C-5 position of the histidine ring irrespective of the fact that iodo is not a particularly preferable halogen to carry in a drug molecule. Hydrophobicity enforced good in-vitro activities is not necessarily a good strategy.
3) there is no attempt to provide a comparative SAR involving other halogens to understand the significance of the iodo group at the C-5 position.
4) there is no attempt to optimize the activities of these analogs by introducing other membrane disrupting groups at the C-5 position other than iodine. Since they have an iodo group there they can easily introduce varieties of lipophilic groups by various organometallic couplings.
Therefore, I strongly recommend extending the SAR a little bit more to see if it results in peptides with superior anticryptococcal activities. In my opinion, the activities of the new analogs better or close to the positive control will be preferable.
Reviewer 3 Report
The paper by Jain and colleagues describes the very concise synthesis of a series of histidine-containing dipeptides and their in vitro characterization as antifungal agents. Starting from the previously reported Trp-His dipeptide, compound 1 in Fig. 1, possessing a good anticryptococcal activity, the authors designed and realized 15 dipeptide analogues. The planned structural modifications included (i) the iodination at the histidine C-5 position, (ii) the substitution of the benzyl amide group for a methyl ester at the carboxylic terminus, (iii) the introduction of variously decorated (with electron donating/electron withdrawing, bulky substituents) benzyl groups at the N-1 position of imidazole ring. The 15 Trp-His dipeptide analogues synthesized (compounds 10a-o) and their N-Boc protected precursors (compounds 9a-o) were preliminarily screened as antifungal agents; 8 out of the 15 deprotected dipeptides were also tested for their antibacterial activity. Lastly, just two compounds (10d and 10o) were assayed for their fungicidal action and compared with clinical standards used for the treatment of invasive cryptococcal infections.
The biological evaluation was carried out on the two selected dipeptides 10d and 10o and the mechanism of action is hypothesized on the basis of electron microscopy analyses. On this evidence, the authors concluded that substitution at histidine C-5 with iodo-group influenced the biological activity against C. neoformans with high selectivity and safety profiles and that antifungal activity can be ascribed to the ability to cause fungal cell lysis.
It is the opinion of this reviewer that the experimental work is well done, the proposed mechanism of action is consistent with the experimental data reported and the literature list is complete, however, the following weaknesses can be identified:
1. The synthetic approach leading to the dipeptide collection is very concise, but authors do not comment the efficiency of the various chemical transformations (the chemical yields are missing in Schemes 1 and 2); also, a comparison with the synthesis of the previous reported dipeptide compounds could be interesting for the readers and should be added.
2. At the end of the SAR description, the authors state that the increase of antifungal activity depends on an increase of hydrophobicity in comparison with the parent Trp-His dipeptide 1: how hydrophobicity has been evaluated? This information should be included.
3. Also, the best antifungal dipeptides (10d and 10o) are as potent as the reference compound 1 (Compound 1: IC50 = 2.10 mg/mL, MIC = 3.81 mg/mL; Compound 10d: IC50 = 2.20 mg/mL, MIC = 4.01 mg/mL); therefore, no substantial improvement in antifungal activity was obtained: the authors should include in Table 2 the data of compound 1 for the sake of comparison and critically comment these results.
4. The use and style of English language throughout the text is quite lacking: it is recommended to revise the manuscript with the help of a native speaker.
Overall, this referee thinks that the manuscript merits publication in Molecules, provided that the above mentioned points of weakness are properly addressed and an accurate revision of the manuscript is done.
Round 2
Reviewer 1 Report
The authors state in the repsonse that peptides are composed of amino acids, which are intrinsic part of body and therefore, lack such side effects and microbes are less prone to develop resistance against them.
However, antimicrobial peptides form pores and disrupts the lipid bilayer of membranes. Side effects may happen even being pepides.
It is true that peptides are degradated to aminoacids in the body, however if this happends peptides would not reach the target cells. Therefore, this is something to be solved in the research area.
The control experiments of combined treatment (comments 2) are mandatory, since this is the interesting point of the manuscript.
Reviewer 2 Report
I am satisfied with the authors response to my concerns. I would advise to add a section in the SAR discussion on the direction of future studies. That would largely eliminate any confusion in reader's mind.
